# Machine Learning Undercounts Reproductive Organs on Herbarium Specimens but Accurately Derives Their Quantitative Phenological Status: A Case Study of *Streptanthus tortuosus*

**DOI:** 10.3390/plants10112471

**Published:** 2021-11-16

**Authors:** Natalie L. R. Love, Pierre Bonnet, Hervé Goëau, Alexis Joly, Susan J. Mazer

**Affiliations:** 1Department of Ecology, Evolution, and Marine Biology, University of California, Santa Barbara, CA 93106, USA; sjmazer@ucsb.edu; 2Biological Sciences Department, California Polytechnic State University, San Luis Obispo, CA 93407, USA; 3Botany and Modeling of Plant Architecture and Vegetation (AMAP), French Agricultural Research Centre for International Development (CIRAD), French National Centre for Scientific Research (CNRS), French National Institute for Agriculture, Food and Environment (INRAE), Research Institute for Development (IRD), University of Montpellier, 34398 Montpellier, France; pierre.bonnet@cirad.fr (P.B.); herve.goeau@cirad.fr (H.G.); 4ZENITH Team, Laboratory of Informatics, Robotics and Microelectronics-Joint Research Unit, Institut National de Recherche en Informatique et en Automatique (INRIA) Sophia-Antipolis, CEDEX 5, 34095 Montpellier, France; alexis.joly@inria.fr

**Keywords:** regional convolutional neural network, object detection, deep learning, visual data classification, climate change, natural history collections, phenological stage annotation, flowering time, phenological shift, phenology

## Abstract

Machine learning (ML) can accelerate the extraction of phenological data from herbarium specimens; however, no studies have assessed whether ML-derived phenological data can be used reliably to evaluate ecological patterns. In this study, 709 herbarium specimens representing a widespread annual herb, *Streptanthus tortuosus,* were scored both manually by human observers and by a mask R-CNN object detection model to (1) evaluate the concordance between ML and manually-derived phenological data and (2) determine whether ML-derived data can be used to reliably assess phenological patterns. The ML model generally underestimated the number of reproductive structures present on each specimen; however, when these counts were used to provide a quantitative estimate of the phenological stage of plants on a given sheet (i.e., the phenological index or PI), the ML and manually-derived PI’s were highly concordant. Moreover, herbarium specimen age had no effect on the estimated PI of a given sheet. Finally, including ML-derived PIs as predictor variables in phenological models produced estimates of the phenological sensitivity of this species to climate, temporal shifts in flowering time, and the rate of phenological progression that are indistinguishable from those produced by models based on data provided by human observers. This study demonstrates that phenological data extracted using machine learning can be used reliably to estimate the phenological stage of herbarium specimens and to detect phenological patterns.

## 1. Introduction

Within and among plant species, the study of phenological traits such as the timing of bud break or flowering can provide key insights into how species respond to climate [1]. Measuring the relationship between phenology and local environmental conditions enables us to evaluate the sensitivity of phenological events to changes in temperature, precipitation, and other climatic parameters as well as to predict future phenological shifts in response to projected climate change [2]. Plant phenology mediates complex species interactions (e.g., plant–plant, plant–herbivore, plant–pollinator; [3,4,5]); therefore, climate change-induced disruptions to plant phenology may have cascading effects on the functions and persistence of ecosystems. While previous studies have documented a wide range of phenological responses to climate and climate change [6,7], our understanding of these responses in many taxa and ecosystems remains incomplete. This gap limits our ability to make broad scale predictions of the ecosystem-wide impacts from climate change [8,9].

Natural history collections, such as herbarium specimens, provide a long temporal record of phenology for hundreds of thousands of taxa globally and offer a data-rich resource with which to fill this gap [6]. Moreover, herbarium specimens can be used to track phenological responses to climate and climate change [10,11,12]. With large-scale efforts to digitize and image herbarium records, millions of herbarium records are now available via large data aggregators, e.g., GBIF (https://www.gbif.org/ (accessed on 1 October 2020))and iDigBio (https://www.idigbio.org/ (accessed on 1 October 2020)), to advance phenological research.

The majority of herbarium specimens include the name of the sampled species (although the scientific name may not be up to date), the precise date of collection, and a text description of the location at which the specimen was collected, which may include a note indicating whether or not the specimen was collected in flower. A smaller fraction of herbarium specimens have also been georeferenced to provide the latitude and longitude of the site of collection (including estimates of the spatial uncertainty of the geographic coordinates, which may range from 0–25 km). These attributes of specimens, when available, are routinely included in electronic databases provided by the herbaria that archive the physical specimens. By contrast, electronic records of herbarium specimens rarely include reports of the numbers of flower buds, open flowers, developing ovaries, and full-sized or mature fruits. Previous work by Love et al. [13], Love and Mazer [14], and Mazer et al. [15] found that analyses that include a quantitative estimate of the phenological status of herbarium specimens—ranging from those bearing only flower buds (i.e., when the plant is collected at the beginning of a reproductive cycle) to those bearing only ripe fruits (i.e., when the plant is collected at the end of an annual or irregular reproductive cycle)—as a predictor variable in models designed to detect the effect of local climate on flowering date can improve their predictive capacity. In addition, this information can be used to estimate the rate at which a species progresses phenologically from bud through fruit production [13]. However, scoring specimens to record the numbers of different types of reproductive organs in order to estimate quantitatively their phenological status requires substantial human investment. Although citizen scientists can provide valuable contributions to the scoring process [16], it remains a challenging task that will be difficult to implement at large scales.

Recently, machine learning techniques—specifically, convolutional neural networks (CNN)—have been used to harvest accurate phenological data from imaged herbarium specimens [17,18,19], and represents a promising approach for large-scale and automated phenological data extraction [20]. For example, Lorieul et al. [17] used CNN to distinguish fertile from non-fertile specimens (i.e., coarse-scale or first-order scoring; [21]) of thousands of species with an accuracy of 96.3% (percent of specimens correctly classified as fertile or non-fertile). Their algorithm was also used for finer-scale (i.e., second-order scoring or scoring the presence/absence of flowers and fruits) scoring and demonstrated an accuracy of 84.3% and 80.5% for flower and fruit detection, respectively.

Machine learning has also been used successfully to detect and count individual reproductive structures on herbarium sheets (i.e., third-order scoring) using instance segmentation (mask R-CNN or regional CNN; [18,19]). Goëau et al. [19] trained a model using only 21 manually-scored herbarium sheets bearing a total of 279 buds, 349 open flowers, 196 developing ovaries, and 212 full-sized fruits that resulted in accurate estimates of the number of reproductive structures present on a given herbarium sheet (77.9% accuracy); however, the level of accuracy depended on the type of structure being detected (i.e., flowers vs. fruits). While these studies demonstrate that machine learning can be used to automate scoring, no studies have assessed whether machine learning-derived phenological data can be used reliably to evaluate ecological patterns such as estimating the phenological sensitivity of flowering date to climate or estimating phenological shifts in response to contemporary climate change (but see [13] for a preliminary study).

In this study, we use a set of 709 herbarium specimens of *Streptanthus tortuosus* Kellogg (Brassicaceae) that were scored both manually (by human observers) and by using mask R-CNN to assess the concordance between human- and machine-learning-derived phenological data and to evaluate the use of automated phenological scoring in phenological research. Specifically, we addressed the following questions:Can machine learning be used to obtain counts of reproductive organs on herbarium specimens that match those recorded by human observers?Do distinct organ types (e.g., flower buds, open flowers, or fruits) differ with respect to the degree to which machine-learned counts match those recorded by human observers?When counts obtained via machine learning are used to estimate the phenological index (PI)—a *quantitative metric* of the phenological status of an herbarium specimen that reflects the proportions of different types of reproductive organs (Love et al., 2019)—does the machine-generated PI match the PI calculated from counts reported by human observers?Does the year of specimen collection affect counting error or estimation of the phenological status? For example, does the fading of floral pigments over time make it more difficult for machine learning algorithms to distinguish between buds and open flowers as herbarium specimens age?When the phenological status (PI) of specimens generated by machine learning is used to construct models that provide estimates of the rate of temporal shifts in flowering date, the sensitivity to climate, and the rate of phenological progression, do these estimates match those based on data derived from human observers?

Despite the potential for errors in human-derived counts of reproductive structures, in the current study we assume that human-counted values are more correct than machine-learning counted values. We treat human-observed values as the true values against which we compare values derived from machine learning to assess its accuracy in extracting phenological data from herbarium specimens.

## 2. Results

### 2.1. Herbarium Data

Our final testing dataset consisted of 709 georeferenced herbarium specimens that were scored both manually and via the machine-learning model. These specimens’ collection dates spanned a 112-year period (1902–2013) and the sites of collection were distributed throughout S. tortuosus’ geographic range (Appendix A). The mean DOY (day of year of specimen collection between 1 and 365) among all specimens was day 182 (1 July; SD = 35.2, range: 76–256). The spring mean maximum daily temperature (spring T_max_) and cumulative winter precipitation experienced at each collection site during the year of specimen collection ranged from 1.4–24.6 °C (SD = 4.8 °C) and 88–2167 mm (SD = 357.8 mm), respectively.

The 21 herbarium specimens used to train the mask R-CNN model were collected between 1899 and 1983. The number of reproductive organs per sheet depended on the type of organ, as follows: buds (range: 0–78; SD = 18), flowers (range: 0–70; SD = 18), immature fruits (range: 0–30; SD = 9), and mature fruits (range: 0–32; SD: 12). The manually-derived PI of the training set ranged from 1.12 to 3.96 (Appendix A).

### 2.2. Phenological Scoring: Manual vs. Machine Learning

The machine learning model generally under-estimated the number of structures relative to manual counts (i.e., human observers) across all categories (buds, flowers, immature fruits, and mature fruits; Figure 1 and Figure 2); however, the degree to which the model under-estimated counts depended on the type of structure being detected. Mature fruits showed the strongest concordance between predicted (machine learning-derived) vs. manual counts (r = 0.75) while flowers showed the weakest (r = 0.57; Figure 1). The slopes of the linear regressions between predicted vs. manual counts indicated that, on average, the algorithm detected 1 bud for every 6.2 manually-counted buds and 1 mature fruit for every 3.6 manually counted mature fruits present on each sheet (buds = 0.16 ± 0.007 predicted per manually-counted bud, *p* < 0.0001; mature fruits = 0.28 ± 0.009 predicted per manually-counted mature fruit, *p* < 0.0001; Appendix A). These slope estimates ranged from 0.16 ± 0.007 buds to 0.30 ± 0.01 immature fruits predicted per manually-counted structure (Appendix A). The regression R^2^ for all models ranged from 0.32 (flowers) to 0.56 (mature fruits; Appendix A). For all organ types, there were fewer counting errors when fewer organs (or overlapping organs) were present on the sheet (Figure 2 and Figure 3; Table 1).

Despite underestimation across all classes, there was high concordance between machine learning-derived estimates of the PI and those derived from manual counts (r = 0.91; Figure 4). The slope of the linear regression between machine-learning- vs. manually-derived PI values for each record also indicates high concordance (slope = 0.91 ± 0.01, *p* < 0.0001, R^2^ = 0.82; Figure 4, Appendix A).

### 2.3. Effect of Year of Specimen Collection on Counting Error

The year of specimen collection had a very small effect on the algorithm’s ability to detect and count some classes of reproductive structures, but had no effect on the estimation of the PI (Table 2). As specimens aged, the machine learning algorithm for counting most organ types became less accurate. The absolute value of counting error (the deviation between human- and machine-counted organs) for flowers, immature fruits, and mature fruits increased with specimen age (coefficient estimates: flowers = −0.003 ± 0.001 errors/year; immature fruits = −0.004 ± 0.001 errors/year, mature fruits = −0.004 ± 0.002 errors/year; Table 2). For example, specimens collected in 1900 have about 0.3 more counting errors on average (representing either an over- or underestimation of 0.3 flowers per sheet) than specimens collected in 2000. Moreover, the explanatory power of these models was very low (R^2^ < 0.01; Table 2). The year of collection had no effect on the accuracy of the machine-learned estimation of the number of buds per sheet (F_1,707_ = 0.8; *p* = 0.37) or the machine-learning-derived estimation of the PI (F_1,707_ = 0.7; *p* = 0.40; Table 2, Figure 5).

### 2.4. Summary and Comparison of Models Constructed with Manually vs. Machine Learning-Derived Phenological Indices

The two models constructed to estimate shifts in flowering date while controlling for either the manually-derived or the machine-learning-derived PI produced similar estimates of the phenological shift through time (Table 3). The models’ 95% confidence intervals for the effect of year on DOY overlapped (model with manually-derived PI: −0.15 to −0.048 days/year, *p* = 0.0002; with machine-learning-derived PI: −0.16 to −0.058 days/year, *p* < 0.0001; Table 3). Both models estimated that the flowering date of *S. tortuosus* advanced between 4.8–16 days during the past century. Moreover, the PI accounted for a similar amount of variance in DOY regardless of whether it was calculated using manually-derived counts (partial R^2^ = 0.27) or machine-learning-derived counts (partial R^2^ = 0.27) of reproductive structures.

The two phenoclimatic models constructed to estimate the sensitivity of *S. tortuosus* to climate while controlling for either the manually-derived or machine-learning-derived PI produced similar estimates of sensitivity to both spring Tmax and winter PPT (Table 4). The 95% confidence intervals largely overlap for both the estimated sensitivity to temperature (model with manually-derived PI: −5.6 to −5.0 days/°C, *p* < 0.0001; with machine-learning-derived PI: −5.7 to −5.1 days/°C, *p* < 0.0001; Table 4) and precipitation (both models: 0.3 to 1.1 days/100 mm, *p* = 0.0002; Table 4). These phenoclimatic models indicate that *S. tortuosus* flowers earlier during relatively warm and dry years.

Both sets of models produced similar estimates of the rate of phenological progression (days/PI unit), regardless of whether the PI was estimated using manually- or machine-learning-derived counts (Figure 6). Among all models, the 95% confidence intervals for the effect of PI on DOY overlap (Table 3 and Table 4, Figure 6). All models estimated that *S. tortuosus* progressed through one “unit” of the PI in 14.6 to 19.4 days. In other words, given the four reproductive stages recorded here (buds, open flowers, developing ovaries, and ripe fruits) it takes individual plants 43.7–58.1 days to progress from the first stage of reproduction (the production of flower buds) through the last stage of reproduction (the completion of fruit ripening).

## 3. Discussion

The mask R-CNN models trained by [19] and applied in the current study to a dataset of 709 herbarium specimens of *S. tortuosus* generally underestimated the number of reproductive structures present on each specimen; however, when these predicted counts were used to estimate the phenological stage of plants on a given sheet (i.e., the phenological index, or PI), the machine-learning-derived and manually-derived PI’s were highly concordant. We also found that scoring older herbarium specimens resulted in slightly more counting errors than more recently collected specimens for some classes of reproductive structures, but specimen age had no effect on estimating the PI of a given sheet.

Finally, we found that machine-learning-derived estimates of the PI could be used in models to accurately evaluate phenological patterns and sensitivity to local climatic conditions. When the machine-learning-derived vs. manually-derived PI’s were used to construct phenological models, both models produced similar estimates of the phenological shift over time, phenological sensitivity to climate, and rates of phenological progression. These results are discussed in detail below.

### 3.1. Mask R-CNN Model Underestimates Organ Counts but Results in High Concordance between Manual- vs. Machine-Learning-Derived Phenological Indices (PIs)

Use of machine learning in phenological scoring of herbarium specimens has recently advanced from lower-order scoring (e.g., presence/absence of reproductive structures; ref. [17]) to higher-order scoring, specifically the detection and counting of individual reproductive structures including buds, flowers, and fruits [18,19]. In contrast to first-order scoring, counting individual reproductive structures provides data that are more valuable for ecological research [20,21]. For example, counts of full-sized fruits can be used to estimate and study the drivers of reproductive output and plant fitness. Moreover, counts provide more insight into the phenological status of a specimen and can be used to estimate a specimen’s phenological index—a quantitative measure of the phenological status of an herbarium specimen. The PI can be included as a predictor variable in phenological models to control statistically for variation in status among specimens when designing models to examine the factors influencing specimen collection date [13,14].

In the current study, the machine-learning algorithm generally underestimated the true number of reproductive structures present on each herbarium specimen of *S. tortuosus* (as counted manually by human scorers), and the number of counting errors was greater and more variable when more organs were present on a given sheet (Figure 1 and Figure 2). This underestimation was more evident for counts of buds and flowers than for immature and mature fruits (Table 1). Davis et al. [18] used a similar machine-learning approach to detect and count reproductive structures on over 3000 herbarium specimens representing six common wildflower species native to the eastern United States. In contrast to the current study, their models produced more accurate counts of flowers and fruits than of buds. In their study, they found that detection and counting of buds was difficult for two reasons. First, very few training specimens (~10%) had buds represented on plants. Second, buds were smaller in size, and their visual appearance was less distinct than either flowers or fruits. Their results suggest that the accuracy of counting various classes of reproductive structures may vary among species and depends on the morphological distinctiveness among classes.

In the current study, the underestimation of organs among all classes may be due to two factors. First, as more reproductive structures are present on an individual sheet, they tend to overlap and become crowded in addition to other occlusions related to the overlapping of many stems or the presence of tape, which makes their detection more prone to error (Figure 3). This may be especially true for buds, which tend to occur in dense clusters at branch tips. Second, specimens in the training dataset had fewer reproductive structures than those in the test dataset (Appendix A). Because annotating hundreds of reproductive structures for training datasets is very time intensive, we chose to annotate specimens with a moderate number of structures per sheet and those with relatively few overlapping structures for which human observers could provide accurate counts. Thus, test specimens with more reproductive structures than those used to train the model were more prone to counting error when scored. A more representative sample for training (in terms of number of organs per sheet, degree of organ overlap, etc.) may have resulted in more accurate counts of reproductive structures but would have required a substantial time investment. Future researchers will have to balance time investment in annotating training specimens with the goals of the study, because relative abundance rather than exact counts of reproductive structures may be sufficient to meet the goals of the project, as was the case for the study presented here and discussed below.

Although the machine-learning model generally underestimated the number of reproductive structures present on specimens, when predicted counts were used to calculate the phenological index (PI), they were highly concordant with the manually-derived PIs (Figure 4). Because estimating the PI relies on the proportions of reproductive structures in each class rather than on the precise counts, the machine-learning-derived PIs were similar to those derived from manual counts (Figure 4).

The current study demonstrates that machine-learning-derived counts of reproductive organs using an object detection approach can be used to accurately estimate a quantitative measure of a herbarium specimen’s phenological status; however, this attribute can be estimated using other machine-learning approaches as well. In contrast to this study, Lorieul et al. [17] used a global visual or “glance” machine-learning approach (based on a ResNet50 deep-learning model) to quantify phenological status using an image classification task in which each of the targeted visual classes was one of nine coded phenophases defined by the relative proportions of heads in bud, flower, or fruit present on individual herbarium specimens of the Asteraceae. For example, a specimen with no heads in bud and 50% of heads in flower and fruit was given a phenophase score of seven, indicating that it was relatively late in its phenological progression [22]. This global visual approach was evaluated using 20,371 herbarium specimens representing 139 Asteraceae species and aimed to classify each specimen into one of the nine distinct phenophases. The classification accuracy among the phenophases varied from 8.6% to 78.9%, with phenophase nine (representing 100% heads in fruit) being the most accurately classified phenophase. This study demonstrates that counting individual reproductive structures is not always necessary for the automated quantification of phenological status. This result may be relevant for studies where performing an object segmentation task is impractical. For example, a global visual approach may be especially useful when scoring species with compound reproductive structures (e.g., heads in the Asteraceae family) where counting individual organs is challenging.

### 3.2. Low Impact of the Specimen Age on Counting Error

Herbarium specimens have recently emerged as reliable sources with which to detect and track phenological change through time [6,7,11]. As such, older specimens, specifically those collected before the onset of contemporary climate change, are critical for these types of studies. However, no studies have been designed to test whether older specimens can be reliably scored by machine learning. In this study, we found that the specimen collection year had little effect on the counting error of each class of reproductive structures (Table 2), and no effect on the accuracy of PI estimates (Figure 5).

This is the first study to demonstrate that machine learning can be used to accurately estimate the phenological status of older herbarium specimens. Importantly, our training dataset included specimens collected as early as 1899 (Appendix A). Including a broad range of collection years in training datasets will likely yield more accurate scores for older specimens and prevent any type of temporal bias in scoring that could occur otherwise. Future researchers interested in using herbarium specimens to detect phenological shifts in response to recent climate change should consider including older specimens in their training dataset to capture some of the color and morphological changes that occur as herbarium specimens are stored for long periods of time. These considerations may be vital to producing accurate estimates of phenological shifts using machine-learning-derived scores.

### 3.3. Machine-Learning-Derived Estimates of the Phenological Status Can Be Used to Assess Ecological Patterns

Herbarium records are important sources of phenological data and can be used to reliably assess ecological patterns; however, no previous studies have been designed to test whether machine-learning-derived phenological scores of herbarium records can be used to accurately estimate such patterns. In the current study, we found that including machine-learning-derived PIs as predictor variables in phenological models produce similar estimates of the phenological sensitivity to climate, temporal shifts in flowering time, and the rate of phenological progression (as estimated by the coefficient of the PI or days/PI unit; Table 3 and Table 4, Figure 6) as models based on data provided by human observers. Moreover, the PI accounted for a similar amount of variance in DOY regardless of whether it was estimated by human scorers or by machine-learning models (Table 3 and Table 4).

Our results demonstrate that machine-learning-derived PIs can be reliably used as a predictor variable in phenological models to (1) control statistically for variation among specimens in their phenological status and (2) estimate the rate of phenological progression (Table 3 and Table 4, Figure 6). However, because we found that the machine-learning model generally underestimated the exact number of reproductive organs present on an individual herbarium specimen, phenological scores determined using the relative abundance of organs, rather than exact counts, may be more reliable when using machine learning to score specimens [13,17]. Use of machine learning could accelerate the process of phenological data extraction and scoring, especially if models trained on one species can be transferred to another [18]; however, more work is needed to determine the reliability of model transfer to different species.

### 3.4. Future Directions

Creating masks to train object detection in machine learning models is a time intensive process; however, the ability to transfer models trained on one species to a closely related and morphological species would enhance our capacity to extract phenological data from imaged herbarium specimens. Davis et al. [18] showed that Mask R-CNN models trained on specimen data of one species can be used to accurately detect and count phenological features of a related species. They suggested that morphological similarity between species may be a better guide for transferability success than phylogenetic relatedness, and that researchers should consider using morphological trait databases to estimate model transferability potential. Many *Streptanthus* species have morphologically similar reproductive structures, and the transferability of the model developed in the current study to other members of the genus will likely generate accurate estimates of specimen PIs.

Community science or crowd-based approaches for scoring specimens could also be leveraged to extend the use of machine learning in phenological research [16]. Crowd-based approaches that engage a large community of non-experts to score herbarium specimens are a reliable way to extract phenological data from herbarium specimens while also providing a large amount of vital training data [18]. A large amount of high-quality training data are key to the performance of automated detection models such as mask R-CNN. The implementation of enrichment and validation data derived from automated predictions could quickly increase the diversity and volume of training data. For example, specimens that have been scored by machine learning with high confidence could then be validated by human scorers, which could enrich the training data and increase the diversity of visual contexts on which models are trained and lead to the generation of more robust models. Finally, herbarium specimens scored using an object-detection-based approach generate masks of individual reproductive structures that can be used to measure the shape and size of individual organs (Figure 3). These phenotypic data could then be leveraged to study drivers of variation in shape and size of organs or to construct morphological trait databases (e.g., [23]).

## 4. Materials and Methods

### 4.1. Assembling Herbarium Records and Manual Phenological Scoring

We assembled and manually scored the phenological status of 1138 herbarium records from seven herbaria (CAS, CHSC, DAV, OBI, RSA, SFV, and UCJEPS). We extracted the day of year of specimen collection (DOY) from the collection information on the label of each herbarium specimen. The DOY, which can range from 1 (1 January) to 365 (31 December), is a widely used estimate of flowering date derived from herbarium specimens and is used commonly as a response variable when estimating sensitivity of flowering date to climate and shifts in flowering time in phenological studies [6,11].

Specimens were then scored using the ImageJ scoring protocol and phenophase definitions described in Love et al. [13]. We used the Cell Counter Plug-in available through ImageJ to score each specimen manually by placing a colored marker on each bud, flower, immature fruit, and mature fruit. ImageJ summed the number of markers per category of reproductive structure on each sheet, and saved the x-y coordinates of the marker on each image. These manually placed markers were subsequently used to annotate images for training the Mask R-CNN model (described in detail below and in [19]).

The manual counts were then used to calculate a quantitative estimate of the phenological stage of each herbarium record: the phenological index, or PI [13]. The PI is designed to be included as a predictor variable in regression models to control statistically for variation in phenological status among herbarium records when estimating the sensitivity of flowering date to climate or shifts in flowering time in response to contemporary climate change. To calculate the PI, each class of reproductive organ is assigned an index value from 1–4 that represents the degree of phenological progression (buds = 1, flowers = 2, immature fruits = 3, and mature fruits = 4). The proportion of reproductive structures in each class is then used to calculate the PI per sheet using the following equation:(1)$$\mathrm{Phenological}\;\mathrm{Index}\;\left(\mathrm{PI}\right)=\overset{4}{\underset{i=1}{\sum }}P_{x}i$$
where *P_x_* is the proportion of reproductive units in each class, and *i* represents the index value assigned to each class of reproductive structure. A specimen with a PI of 1 would be bearing only buds, indicating that the specimen was at an early stage of phenological progression when collected. In contrast, a specimen with PI value of 4 would be bearing only mature fruits and would indicate that the specimen was collected at a late reproductive stage. Specimens with values of the PI between 1 and 4 would typically be bearing a combination of at least two of the four classes of reproductive structures used in the current study. Including the PI as a predictor variable in linear models designed to detect the effects of local climate or year on the day of year of specimen collection is useful because the value of PI is highly positively correlated with DOY (controlling for variation in local climate) and therefore contributes significantly to the model R^2^. All else being equal, specimens with low values of the PI were collected earlier in the growing season than those with high values of the PI. In addition, the regression coefficient associated with the PI provides an estimate of how many days it takes for plants to increase their PI by a value of one (see Results section for an example using the data analyzed here).

Next, we georeferenced each specimen by either downloading the latitude and longitude for that record from the California Consortium of Herbaria (CCH2; www.cch2.org (accessed on 3 March 2018)) or using the written description of the collection location to derive coordinates using GEOLocate (www.geo-locate.org (accessed on 3 March 2018)), a platform for georeferencing natural history specimens or electronic records. Each set of coordinates was also associated with an error radius (in meters) that represented the uncertainty of each location. After georeferencing, we excluded records with an error radius greater than 4000 m and duplicate records. This resulted in a set of 709 specimens that were then scored using the trained algorithm.

#### Extracting Climate Data

For each of the georeferenced specimen records, we extracted from ClimateNA [24] two climate variables that are known to be associated with flowering date in *S. tortuosus*: the maximum spring temperature (spring Tmax) and the cumulative winter precipitation (winter PPT) during the year of specimen collection [13,14]. ClimateNA is a data source that downscales gridded PRISM data to scale-free point locations. While it is possible that other climatic parameters may be better at predicting the flowering time of this species, testing all possible variables (and selecting the model of best fit) was beyond the scope of this study; our goal was to select a reasonable model to use in order to compare the performance of models using data obtained from a machine learning algorithm vs. from human observers.

### 4.2. Machine Learning Methods

#### 4.2.1. Training Dataset

We uploaded images of herbarium specimens into COCO Annotator (https://github.com/jsbroks/coco-annotator (accessed on 2 February 2019), a web-based tool for object segmentation), in order to manually draw the full outline of each reproductive structure. This allowed us to capture the full shape of each reproductive organ identified on each specimen. When structures overlapped on the specimen, only the structure in the foreground was annotated, resulting in the exclusion of background structures from the segmentation in that part of the image. A total of 1036 reproductive structures from 21 herbarium specimens were annotated (flower buds (279), flowers (349), immature fruits (196), and mature fruits (212)). This dataset is freely available on the Zenodo platform [25].

#### 4.2.2. CNN Training and Prediction

For this study, we used the model trained by Goëau et al. [19]. This fine-grained detection method is based on the mask R-CNN architecture [26], which has demonstrated its robustness and efficiency in instance segmentation tasks and challenges such as MS COCO (Microsoft Common Objects in Context; [27]). This architecture used Facebook’s mask R-CNN framework [28] implemented with PyTorch [29]. Goëau et al. [19] chose ResNet-50 as the backbone CNN and the Feature Pyramid Networks [30] for instance segmentation. The selection of the number of training iterations was made based on the empirical observation of the model’s training performance. The details of the training hyperparameters are provided in Appendix B, and for a full discussion of the evaluation of the mask R-CNN model used in this study, see Goëau et al. [19].

### 4.3. Model Evaluation and Statistical Analyses

#### 4.3.1. Evaluation of Machine Learning Model Predictions

We evaluated the concordance between manual counts and predicted counts from the machine learning model in two ways [18]. First, we calculated the counting error ($e_{i,k}$) per class of reproductive organ (*k*) on each herbarium sheet (*i*), which is defined as the difference between the number of organs counted manually by human observers (${\hat{c}}_{i,k}$) and the number predicted by the machine learning model ($c_{i,k}$). We used the following equation:(2)$$e_{i,k}={\hat{c}}_{i,k}-c_{i,k}$$
where *k* represents buds, flowers, immature fruits, or mature fruits. Positive counting errors represent cases in which the model overestimates the number of structures per sheet, while negative counting errors represent cases in which the model underestimates the number of structures. We also assessed whether the total number of reproductive structures per sheet affected the degree of counting error by constructing box plots.

Second, we calculated the mean absolute error (*MAE*) per class of reproductive structure, which measures the mean absolute error per class of reproductive structure across all sheets (*N*). We used the following equation:(3)$$MAE=\frac{1}{N}\underset{i}{\sum }\;\underset{k}{\sum }\left|e_{i,k}\right|$$

#### 4.3.2. Statistical Analysis

We calculated the Pearson correlation coefficients between manual- and machine-learning-derived counts to assess their concordance for each class of reproductive organ as well as for the PI. We also fit a linear regression to these relationships. To determine how the number of organs present on each sheet affected the algorithm’s ability to accurately count them, we plotted the distribution (as box plots) of count errors for each of five ranges of reproductive organs present (e.g., 0–9 organs, 10–19 organs, etc.).

To determine whether the year of collection affected the model’s ability to accurately count the number of reproductive structures on each sheet, we constructed one linear model for each of the four organ classes that assessed the effect of year of collection on the absolute value of the counting error. We used the absolute value of counting error in this analysis because we were interested in estimating the effect of year on the overall error rate and not whether the algorithm specifically over- or underestimates the organ count through time. We also used linear regression to assess whether the year of collection affected the estimation of the PI. To improve normality, the absolute value of counting error was log_10_ transformed after adding 1 to each value, and the PI was square root transformed.

We constructed two sets of multivariate linear regression models (with two models each) to determine whether the PI derived from machine learning counts (or predicted counts) could be used to construct models that accurately estimate (relative to the human observations) the rate of temporal shifts in flowering date (set 1) and the sensitivity of flowering date to climate (set 2). Models in each set controlled for variation in phenological status among specimens by including the PI derived from either manual or predicted counts, but otherwise included the same predictors. The first set of models (set 1) assessed the effect of year on the DOY to determine the rate of phenological change (days/year), and included year, PI, elevation, latitude, and longitude as main effects. The second set of models (set 2) assessed the sensitivity of flowering date (DOY) to climate during the year of specimen collection and included PI, cumulative winter precipitation, and spring maximum temperature. Temperature and precipitation have been demonstrated to affect flowering date in *Streptanthus tortuosus* where the date of specimen collection is used as a proxy for flowering date (Love et al., 2019; Love and Mazer, 2021). For both sets of models, we calculated the 95% confidence interval of each coefficient estimate as well as the partial R^2^ for each main effect.

Because both sets of models included the PI as a main effect, this allowed us to compare the rate of phenological progression (days/PI unit) as estimated by the regression coefficient of the PI (whether the PI was estimated from manual or machine-learned counts) among all four models. We used the 95% confidence interval to evaluate whether the estimated rate of phenological progression differed between counting methods (manual vs. machine learning) or among models. All statistical analyses were conducted using R version 4.0.3 [31].

## Figures and Tables

**Figure 1 plants-10-02471-f001:**
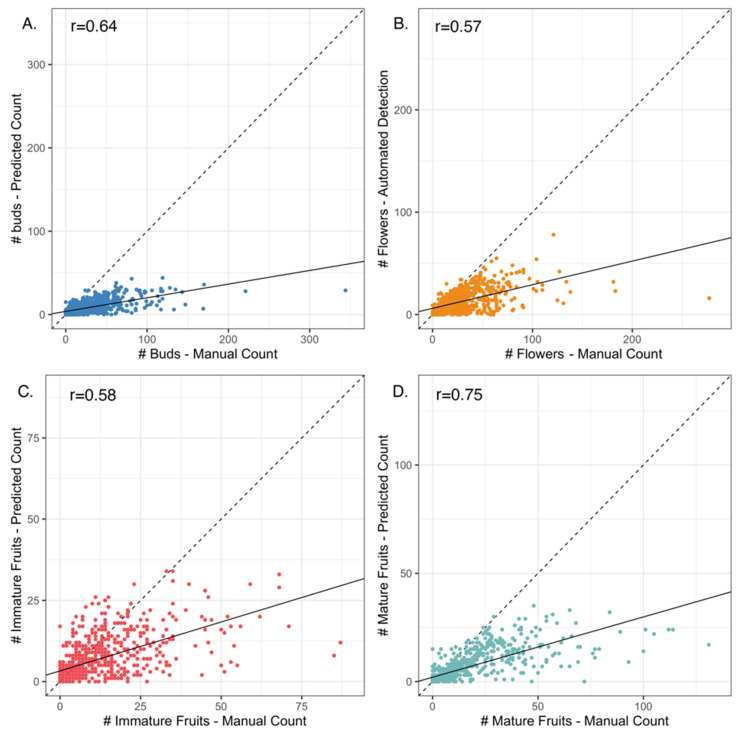
Bi-variate relationship between the number (#) of organs predicted by the machine-learning model vs. those counted manually by human observers for (**A**) buds, (**B**) flowers, (**C**) immature fruits, and (**D**) mature fruits. The dashed diagonal line shows a hypothetical 1:1 relationship where the manual and true counts are the same. The solid line denotes the linear regression between predicted and manual counts. Summaries for each regression can be found in Appendix A.

**Figure 2 plants-10-02471-f002:**
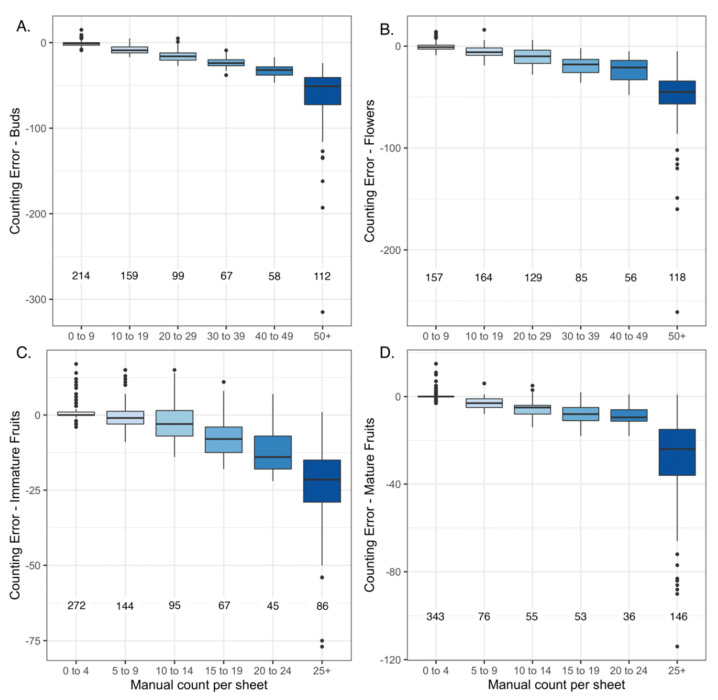
Box plots of counting error vs. number of reproductive units present on each herbarium specimen (as estimated manually by human observers) for (**A**) buds, (**B**) flowers, (**C**) immature fruits, and (**D**) mature fruits. The numbers below each box plot denote the number of herbarium specimens represented in each box plot.

**Figure 3 plants-10-02471-f003:**
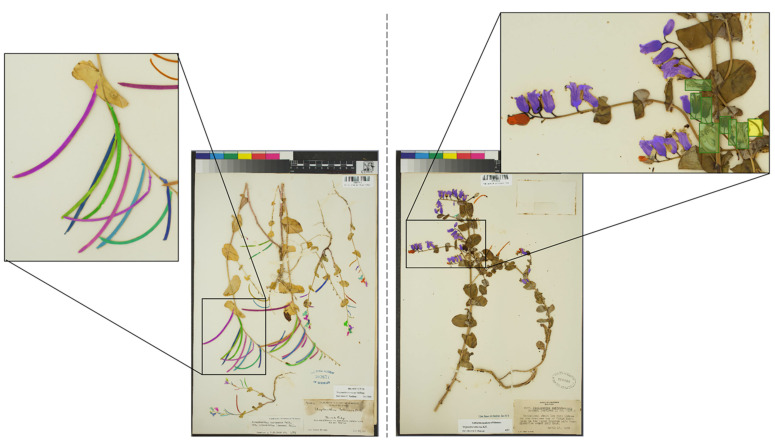
Examples of manually produced masks used to train the recognition model. In the specimen on the left, several mature fruits overlap, so only the structures in the foreground were annotated in this case, resulting in the exclusion of background structures from the segmentation in that part of the image. The specimen on the right displays examples of predicted masks (buds in red, flowers in purple, immature fruits in orange, mature fruits in blue) on a test specimen. The organs that were not detected (flowers in green rectangles and immature fruit in a yellow rectangle) are either very largely degraded, or partially hidden behind stems or leaf lamina.

**Figure 4 plants-10-02471-f004:**
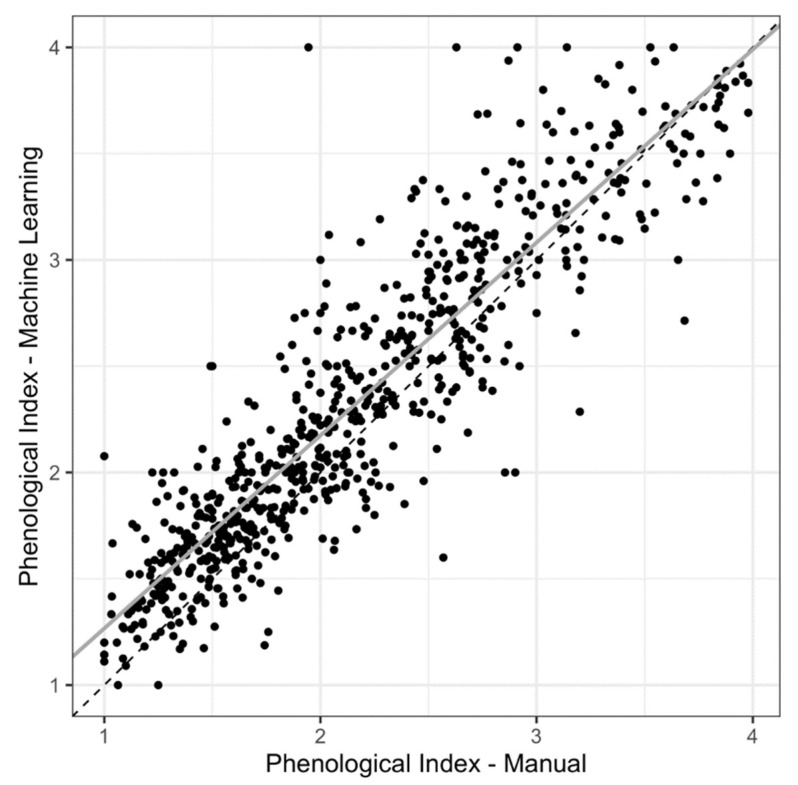
Bivariate plot showing the relationship between the phenological index (PI) as estimated using predicted counts vs. manual counts (counted by human observers). The solid gray line denotes the linear regression between the two estimates of PI. The dashed black line shows a hypothetical 1:1 relationship where the two estimates result in the same PI value. Summaries for the regression can be found in Appendix A.

**Figure 5 plants-10-02471-f005:**
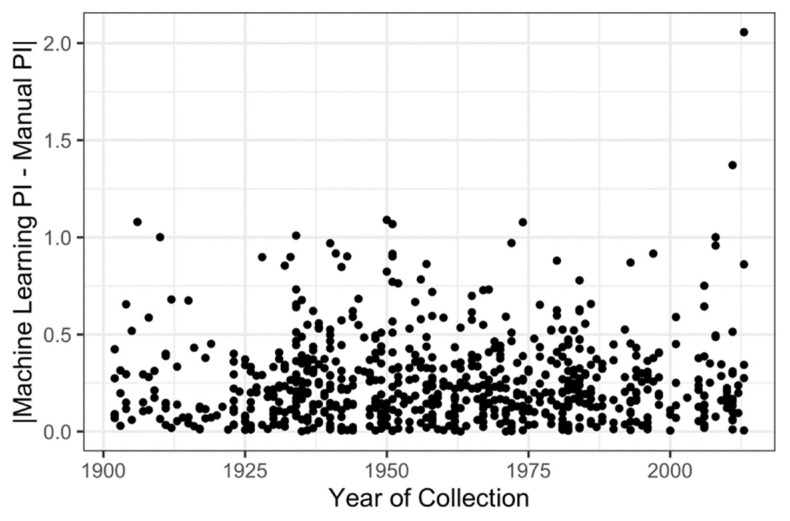
Bivariate plot showing the relationship between the year of specimen collection (i.e., specimen age) and the absolute difference between the machine-learning-derived estimate of the phenological index (PI) and the manually-derived (from human observers) estimate of the PI, or PI estimation error. There is no relationship between estimation error and the year of specimen collection.

**Figure 6 plants-10-02471-f006:**
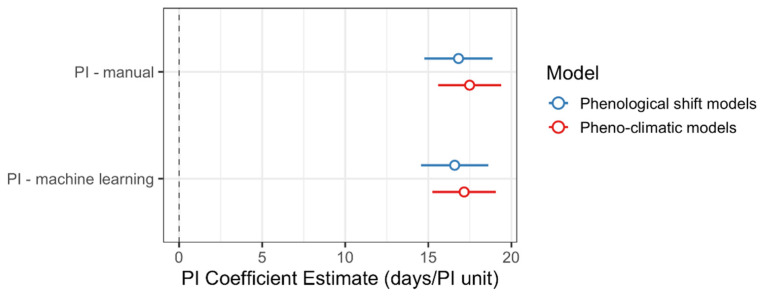
Estimates for the rate of phenological progression (in days/phenological index unit) of *Streptanthus tortuosus* as measured by the coefficient of the phenological index (PI) from linear regression models designed to detect temporal shifts in flowering date and sensitivity of flowering date to climate. The 95% confidence interval for each estimate can be found in Table 3 and Table 4.

**Table 1 plants-10-02471-t001:** Summary statistics showing the numbers of reproductive structures in each class present on each sheet as determined by human observers (i.e., manually-derived counts). The mean average error (MAE) was calculated and used to evaluate the performance of the machine learning model. The MAE measures the mean absolute error per class of reproductive structure across all sheets.

Reproductive Structure	Number per Sheet (Range)	Mean ± SD per Sheet	MAE
Buds	0–344	27.4 ± 30.6	19.7
lFowers	0–227	28.5 ± 26.4	16.6
Immature fruits	0–87	11.0 ± 12.7	6.5
Mature fruits	0–131	12.5 ± 19.3	8.3

**Table 2 plants-10-02471-t002:** Summaries of the linear regressions designed to detect the effect of year of specimen collection (i.e., specimen age) on the absolute value of counting error for each class of reproductive structure and on the estimation of the phenological index (absolute value of the difference between machine-learning-derived and manually-derived phenological index or PI). Bolded values represent significant relationships at *p* < 0.05.

Reproductive Structure	Slope Estimate ± SE	*p*-Value	R^2^
Buds	−0.0015 ± 0.002	0.3699	0.001
Flowers	**−0.0036 ± 0.001**	**0.0204**	0.008
Immature fruits	**−0.0037 ± 0.001**	**0.0095**	0.008
Mature fruits	**−0.0045 ± 0.002**	**0.0083**	0.008
phenological index	0.00024 ± 0.0003	0.3979	0.001

**Table 3 plants-10-02471-t003:** Summary of the multiple linear regression designed to detect temporal shifts in flowering date while controlling for the phenological status as estimated by the (a) manually-derived or (b) machine-learning-derived phenological index (PI). Bold values represent significant effects at *p* < 0.05.

a.						
Independent Variable	Estimate	SE	95% CI	*t*-Ratio	*p*-Value	Partial R^2^
Intercept	216.77	133.67	-	1.62	0.1053	-
Year of collection	−0.1	0.027	−0.15 to −0.048	−3.73	**0.0002**	0.02
PI (manual)	16.82	1.04	14.77 to 18.87	16.11	**<0.0001**	0.27
Elevation	0.03	0.0011	0.027 to 0.032	27.33	**<0.0001**	0.52
Latitude	5.18	1.14	2.94 to 7.41	4.54	**<0.0001**	0.03
Longitude	1.06	1.29	-	0.82	0.412	<0.01
					Full Model R^2^	0.7
**b.**						
**Independent Variable**	**Estimate**	**SE**	**95% CI**	***t*-Ratio**	***p*-Value**	**Partial R^2^**
Intercept	134.36	133.13	-	1.01	0.3132	-
Year of collection	−0.11	0.027	−0.16 to −0.058	−4.14	**<0.0001**	0.02
PI (machine learning)	16.59	1.03	14.56 to 18.6	16.1	**<0.0001**	0.27
Elevation	0.031	0.0011	0.028 to 0.033	28.54	**<0.0001**	0.54
Latitude	4.29	1.14	2.06 to 6.51	3.78	**0.0002**	0.02
Longitude	−0.056	1.28	-	−0.043	0.9654	<0.01
					Full Model R^2^	0.7

**Table 4 plants-10-02471-t004:** Summary of the multiple linear regressions designed to estimate the sensitivity of flowering date to climate during the year of specimen collection while controlling for the phenological status as estimated by the (a) manually-derived or (b) machine-learning-derived phenological index (PI). Bold values represent significant effects at *p* < 0.05.

a.						
Independent Variable	Estimate	SE	95% CI	*t*-Ratio	*p*-Value	Partial R^2^
Intercept	205.58	3.33	-	61.65	**<0.0001**	-
PI (manual)	17.49	0.97	15.59 to 19.38	18.09	**<0.0001**	0.32
Winter PPT	0.0072	0.0019	0.0034 to 0.011	3.74	**0.0002**	0.02
Spring Tmax	−5.28	0.14	−5.56 to −5.00	−36.71	**<0.0001**	0.66
					Full Model R^2^	0.73
**b.**						
**Independent Variable**	**Estimate**	**SE**	**95% CI**	***t*-Ratio**	***p*-Value**	**Partial R^2^**
Intercept	204.75	3.41	-	59.96	**<0.0001**	-
PI (machine learning)	17.16	0.97	15.25 to 19.06	17.7	**<0.0001**	0.31
Winter PPT	0.0072	0.0019	0.0034 to 0.011	3.74	**0.0002**	0.02
Spring Tmax	−5.38	0.14	−5.66 to −5.09	−37.27	**<0.0001**	0.66
					Full Model R^2^	0.73

## Data Availability

All data used to conduct this study can be found at https://doi.org/10.5281/zenodo.5574675 (accessed on 18 October 2021).

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
