# Peer review of "Machine Learning Undercounts Reproductive Organs on Herbarium Specimens but Accurately Derives Their Quantitative Phenological Status: A Case Study of Streptanthus tortuosus"

_plants, 2021, doi:10.3390/plants10112471_

Round 1

Reviewer 1 Report

Comments for the authors

Manuscript ID: plants-1450341

Title: Machine learning undercounts reproductive organs on herbarium specimens but accurately derives their quantitative phenological status: a case study of Streptanthus tortuosus.

In this manuscript, Love and coauthors present a very interesting use of machine learning applied to known the phenology of the species Streptanthus tortuosus using herbarium specimens. I enjoyed reading this manuscript, and I think it makes a timely an important contribution to the phenological data from herbarium material. The present manuscript has the potential to be a good contribution to the literature and would be of interest to a broad audience.

I just have a few comments to improve the manuscript.

Page 4, line 132. What does DOY mean?

Page 4, line 132. Map references and a legend for the figure are missing. Countries? North hemisphere?

The methods section does not have the same order of appearance as the objectives as well as the statistical analyzes. The year of the specimen is explained before PI, and they have the inverse order in the objectives of the manuscript.

I believe that this can become a strong and widely read Plant paper.

Reviewer 2 Report

The article has been written to a high standard, and the abstract reads really good.

Visual results from mask R-CNN is valuable in showing how different the ML results are from the ground truth (manually labelled).

“Mask R-CNN model underestimates organ counts but results in high concordance between manual vs. machine-learning derived phenological indices (PIs)”.

  • Perhaps, another detector (YOLOv3, Single Shot Detector, Retina-Net or RefineDet) could provide better results. It is worth investigating them in the future.
  • Another issue could be from the backbone used (ResNet50). It is worth investigating which backbone network could work the best for your dataset. Perhaps, one with a few layers than the ResNet50 or more layers than it.
  • Note that the ML model is learning from manually labelled ground truth images, so may not perform better than it.
  • You could have produced and reported a graph of the training loss versus the epochs (learning curve), which is a way to know if the model is working well. Perhaps, the backbone has too many layers, and it will help you establish if your model is overfitting or not.

Metrics:

“Second, we calculated the mean absolute error (MAE) per class of reproductive structure, which measures the mean absolute error per class of reproductive structure across all sheets (N). We used the following equation:”

You have mentioned and given the formula for MAE per class, but you may need to report the results (MAE) per class.

It is worth evaluating the accuracy of your segmentation model. Mean Intersection-Over-Union (Mean IOU) is a metric grounded in literature for this purpose. They will help establish if your model is segmentation correctly. Again, if the model is not working well, it may reinforce why the Mask R-CNN model could be underestimating organ counts.

Reviewer 3 Report

The manuscript entitled “Machine learning undercounts reproductive organs on herbarium specimens but accurately derives their quantitative phenological status: a case study of Streptanthus tortuosus” presents a study using a set of 709 herbarium specimens of Streptanthus tortuosus Kellogg (Brassicaceae) that were scored both manually (by human observers) and by using mask R-CNN to assess the concordance between human- and machine-learning derived phenological data and to evaluate the use of automated phenological scoring in phenological research.

The work is properly organized and structured, with clear results, easy to follow and logically explained, with proper conclusions (avoiding any speculation) based on the data obtained.  To my understanding the methodology applied is adequate and complete.

In conclusion, the results presented by the authors are of scientific relevance, with interesting conclusions so, I recommend its publication in Plants.

Author Response

The authors would like to thank reviewer 3 for their kind comments regarding our work.